# The menstrual cycle and the COVID-19 pandemic

Anita Mitra[1,2], Jan Y. Verbakel[3,4], Lorraine S. Kasaven[1,5]*, Menelaos Tzafetas[1,2], Karen Grewal[1,2], Benjamin Jones[1,2], Phillip R. Bennett[1,2], Maria Kyrgiou[1,2], Srdjan Saso[1,2]

1 Institute of Reproductive and Developmental Biology, Imperial College London, London, United Kingdom, 2 Department of Obstetrics & Gynaecology, Imperial College NHS Trust, London, United Kingdom, 3 Department of Public Health and Primary Care, KU Leuven, Leuven, Belgium, 4 Nuffield Department of Primary Care Health Sciences, University of Oxford, Oxford, United Kingdom, 5 Department of Cutrale and Perioperative Ageing Group, Imperial College London, London, United Kingdom

* lk226@doctors.org.uk

## Abstract

### Background

The impact of COVID-19 virus on menstrual cycles in unvaccinated women is limited.

### Objective

To investigate the prevalence of changes to menstrual cycle characteristics, hormonal symptoms and lifestyle changes prior to and during the COVID-19 pandemic.

### Methods

A retrospective online cross-sectional survey completed by social media users between July 2020 to October 2020. Participants were living in the United Kingdom (UK), premenopausal status and, or over 18 years of age.

### Main outcome(s) and measures(s)

The primary outcome was to assess changes to menstrual cycle characteristics during the pandemic following the Checklist for Reporting Results of Internet E-Surveys (CHERRIES). Secondary outcomes included assessment of hormonal and lifestyle changes.

### Results

15,611 social media users completed the survey. Of which, 75% of participants experienced a change in their menstrual cycle, with significantly greater proportions reporting irregular menstrual cycles (P<0·001), bleeding duration more than seven days (P<0·001), longer mean cycle length (P<0·001) and overall bleeding duration (P<0·001). Over half the participants reported worsening of premenstrual symptoms including low mood/depression, anxiety and irritability. When stratified according to COVID-19 infection, there was no significant difference in menstrual cycle changes.

**Data Availability Statement:** The minimal anonymised data set necessary to replicate our study findings is uploaded as a Supporting Information file.

**Funding:** The author(s) received no specific funding for this work.

**Competing interests:** The authors have declared that no competing interests exist.

## Conclusion

The COVID-19 pandemic resulted in considerable variation in menstrual cycle characteristics and hormonal symptoms. This appears to be related to societal and lifestyle changes resulting from the pandemic, rather than to the virus itself. We believe this may have an impact on the individual, as well as national economy, healthcare, and population levels, and therefore suggest this should be taken into consideration by governments, healthcare providers and employers when developing pandemic recovery plans.

## Introduction

COVID-19 infection caused by SARS-CoV-2 resulted in significant mortality and morbidity, as well as long term effects on multiple organs and systems [1]. Following implementation of the COVID-19 vaccination programme, the UK's Medicines and Healthcare Products Regulatory Agency (MHRA) received more than 36,000 yellow card surveillance scheme for adverse drug reactions reports, including menstrual cycle changes and unexpected vaginal discharge within the first 9 months following vaccine roll out [2]. A recent systematic review has since also demonstrated evidence that exposure of the COVID-19 vaccine is related to menstrual abnormalities, including menorrhagia, metrorrhagia and polymenorrhoea [3].

As many large scale COVID-19 studies excluded menstrual related questions, the impact of the virus on unvaccinated women remains questionable. During the peak of the pandemic, many discussions on social media platforms and blogs suggested women experienced changes to their menstrual cycle; with respect to frequency, duration, regularity, volume and increased dysmenorrhoea and worsening premenstrual symptoms (PMS) [4], however, primarily due to speculation from users self-reporting symptoms on social media, rather than from scientific evidence. This may explain why from a sample of 13,128, 525 women of reproductive age, vaccine uptake amongst women aged 18–29, 30–39 and 40–49 was 36.3%, 34.1% and 29.5% respectively [5], with almost two thirds of women remaining unvaccinated as recently as February 2022. Vaccine hesitancy amongst young women is often attributed to concerns regarding chances of future pregnancy [6], with many sourcing healthcare information through digital mediums [7]. It is therefore apparent that further scientific understanding of the impact of the virus on menstrual cycles in unvaccinated women is required, as it may prevent escalation of COVID-19- related reproductive health concerns, which could otherwise put women at risk, should they delay getting vaccinated or seeking medical treatment.

Regularity of menstrual cycles are often deemed a significant marker of health and wellbeing [8], whereby women who experience irregular and long cycles, are at greater risk of premature mortality [9], and absent cycles or oligomenorrhoea are often an indicator of reduced fertility [10]. Understanding the symptoms related to menstruation is important because of the associated economic burdens they pose, through reduced productivity and increased absence from the workplace [10–13]. Particularly as 26% of the global population are women of reproductive age, with many contributing to the workforce [14]. A recent study highlighted the need for further understanding of the impact of COVID-19 on menstrual cycles, so that findings can be used to guide COVID economic recovery plans, by focussing on improving female productivity amongst those suffering from menstrual related conditions [14].

It has been argued that women have been inexplicably affected by the pandemic compared to men, due to the unequal share of responsibility regarding childcare and lack of security with

employment and finances [15]. In addition to the significant male bias with regards to the development of vaccines, medications and diagnostics, the pandemic has highlighted inequalities in research and development dedicated to women's health [16]. There is an increasing demand therefore, for innovation to tackle gender differences, where understanding of menstrual related health problems could help mitigate the impact of the pandemic on reproductive health, whilst also minimising gender based health and social inequalities [14].

The aim of this study is to investigate the prevalence of changes to menstrual cycle characteristics and hormonal symptoms over the course of the COVID-19 pandemic, prior to the roll out of a vaccination programme. In addition, we sought to describe the lifestyle changes taken place in the female population during the peak of the government induced lockdown.

## Materials and methods

We undertook an online cross-sectional survey to quantify changes in menstrual cycle characteristics and hormonal symptoms during the COVID-19 pandemic in the UK, between July 2020 to October 2020, following the Checklist for Reporting Results of Internet E-Surveys (CHERRIES) [17] and the checklist for Strengthening the Reporting of Observational studies in Epidemiology (STROBE) [18].

A cross sectional survey was used as this was deemed the most appropriate method for understanding determinants of health and describing changes in behaviours in a population during the COVID-19 pandemic. It is also an efficacious method of collecting preliminary evidence to plan future studies. The study was approved by Imperial College London Ethics Committee and advertised via social media (Instagram & Twitter), with a link to the study landing page on Qualtrics XM platform (Qualtrics, Provo, UT). The main author of the study advertised the survey through the *short stories* feature, explaining the purpose of carrying out the research and how to complete the survey via their Twitter and Instagram account, which had a combined following of more than 100 thousand members from the general public. This was felt to be the most appropriate platform to both inform and disseminate the survey to members of the public, given the access to a large number of followers.

The survey was open to participants who were UK-based, of reproductive age (18 years or above), premenopausal and to women irrespective of the type of contraception they used. Electronically signed informed consent was required before entering the study. There were no incentives for participation. All responses were anonymous and the IP address was not recorded.

The survey consisted of three sections. Section 1 assessed women's demographics. Section 2 asked questions relating to menstrual cycle characteristics & hormonal symptoms prior to and during the COVID-19 pandemic. This included questions regarding cycle regularity and length, duration of bleeding, presence of dysmenorrhoea, menorrhagia or intermenstrual bleeding (IMB), premenstrual symptoms including abdominal bloating, feeling irritable, fatigue, low mood or anxiety and changes to libido. Section 3 addressed lifestyle changes since the COVID-19 lockdown by asking questions about commuting to work, home-schooling, frequency of exercise and drinking and changes to diet and sleep patterns.

Participants were able to omit questions, review and change their answers before submission and could discontinue or withdraw from the study at any time. Access to participant identifiable information was limited to the co-authors involved in data analysis of the study only.

The primary outcome was to assess changes to menstrual cycle characteristics during the pandemic. Secondary outcomes included assessment of hormonal and lifestyle changes. The COVID-19 pandemic was considered the exposure variable.

## Statistical analysis

McNemars, Chi-squared, paired and unpaired t-tests were performed where appropriate using STATA Version 16.1 (StataCorp LLC, College Station, TX). Multivariate regression analysis was performed using R software Version 4.0.4 (R Project for Statistical Computing). It was not possible to perform sample size or power calculation *a priori*. P value <0.05 was considered statistically significant. Missing data has not been reported on.

For studies carried out on social research, the best practice error margin is considered ±5% (at a 95% confidence level). This assumes that the target sample size is large enough, such that if 50% of the participants were to respond to an answer with 'yes,' we can be 95% confident that the actual figure in the survey would average between 45–55%. Therefore, the optimal sample size required to achieve a ±5% error margin on a population over 1 million is 385 participants [19].

## Results

### Participants

In total, 15,611 participants responded to the online questionnaire and provided information of at least one menstrual cycle element. **Table 1** describes the characteristics of participants.

The mean age was 29.9 years (range 18–55 years), with the majority falling into the 21–30 year age bracket (n = 8,563; 54·9%). Participants were predominantly Caucasian (n = 13,557; 86·8%) and nulliparous (n = 11,891; 76·2%).

Prior to the March 2020 lockdown, 6·1% (n = 930) of participants were actively trying to conceive and the majority (n = 666; 71.6%) continued to do so during. Reasons cited by the 28.4% (n = 264) who stopped trying to conceive, included concern regarding safety of infection to a pregnancy, lack of access to fertility treatment or general healthcare and financial concerns.

A menstrual cycle tracking app/diary/other tracking method was used by 69·3% (n = 10,245) of participants. A change in libido was reported by 57·9% (n = 8,725); of whom two-thirds noted a decrease (n = 5694; 65·2%), and one-third an increase in libido (n = 3031; 34·7%).

Ten percent (n = 1,508) of participants were amenorrhoeic prior to the lockdown. Reasons reported included the use of hormonal contraception (n = 1,014; 78.9%), polycystic ovarian syndrome (PCOS) (n = 83; 6·4%) and cause unknown (n = 49; 3·8%). Bleeding re-started during lockdown in 48·9% (n = 612) of amenorrhoeic participants, with one-third reporting regular cycles. This was most frequently seen in those with amenorrhoea secondary to PCOS (n = 47; 56·6%) and hormonal contraception (n = 486; 48·0%).

### Changes to menstrual cycle and premenstrual symptoms

Overall, 75·8% (n = 11,833) of participants reported a change to at least one aspect of their periods relating to regularity, pain, amount of bleeding or intermenstrual bleeding (IMB). Of the 90% (n = 13,557) of participants who had periods prior to the lockdown, 3.2% (n = 433) became amenorrhoeic. As shown in **Table 1** and **Fig 1**, there was a significant difference (p<0·001) in the proportion of participants reporting regular menstrual cycles before the start of the pandemic, (n = 10463; 84·4%) to during the lockdown (n = 7642; 61·5%).

Furthermore, the number reporting a bleeding duration greater than seven days almost doubled (pre pandemic 5·1% vs during 9·4%; p<0·001) **(Fig 1)**.

There was an association between younger age (per 10 years) and risk of irregular cycles (Odds ratio (OR) 1·34 (95% Confidence interval (CI)) 1·25–1·45) and younger age and bleeding more than 7 days (1·26 (1·14–1·38)) **(Table 2)**. Parity was also associated with risk of

**Table 1. Menstrual cycle characteristics, all participants.**

| | | Pre-lockdown | During lockdown | p-value |
|---|---|---|---|---|
| Regular menstrual cycle (n/N, %) | | 10463/12402 (84·4) | 7642/12430 (61·5) | <0·001 |
| Bleeding duration >7days (n/N, %) | | 623/12304 (5·1) | 1161/12398 (9·4) | <0·001 |
| Duration of bleeding, days (mean, SD) | | 5·10, 1·9 | 5·34, 2·7 | <0·001 |
| Menstrual cycle length, days (mean, SD) | | 29·6, 13·0 | 30·1, 14·0 | <0·001 |
| Painful periods (n/N, %) | | | | |
| | Yes | 6910/11933 (57·9) | Same – 3388/6910 (49·0) | |
| | | | More – 2265/6910 (32·8) | |
| | | | Less – 1257/6910 (18·2) | |
| | No | 5023/11933 (42·1) | Same – 2908/5023 (57·9) | |
| | | | More – 2048/5023 (40·8) | |
| | | | Less – 67/5023 (1·3) | |
| Heavy periods (n/N, %) | | | | |
| | Yes | 5773/11906 (48·5) | Same – 2693/5773 (46·6) | |
| | | | More – 1256/5773 (21·8) | |
| | | | Less – 1824/5773 (31·6) | |
| | No | 6133/11906 (51·5) | Same – 3895/6133 (63·5) | |
| | | | More – 1714/6133 (28·0) | |
| | | | Less – 524/6133 (8·5) | |
| Intermenstrual spotting/bleeding (n/N, %) | | | | |
| | Yes | 2142/13181 (16·3) | Same – 730/2142 (34·1) | |
| | | | More – 604/2142 (28·2) | |
| | | | Less – 808/2142 (37·7) | |
| | No | 11039/13181 (83·7) | Same – 8753/11039 (79·3) | |
| | | | More – 2151/11039 (19·5) | |
| | | | Less – 135/11039 (1·2) | |
| Abdominal bloating (n/N, %) | | | | |
| | Yes | 9372/13300 (70·5) | Same – 5257/9372 (56·1) | |
| | | | More – 3357/9372 (35·8) | |
| | | | Less – 758/9372 (8·1) | |
| | No | 3928/13300 (29·5) | Same – 2645/3928 (67·3) | |
| | | | More – 1242/3928 (31·6) | |
| | | | Less – 41/3928 (1·0) | |
| Irritability (n/N, %) | | | | |
| | Yes | 8980/13230 (67·9) | Same – 3022/8980 (33·7) | |
| | | | More – 5065/8980 (56·4) | |
| | | | Less – 893/8980 (9·9) | |
| | No | 4250/13230 (32·1) | Same – 1927/4250 (45·3) | |
| | | | More – 2271/4250 (53·4) | |
| | | | Less – 52/4250 (1·2) | |
| Fatigue (n/N, %) | | | | |
| | Yes | 8109/13251 (61·2) | Same – 2329/8109 (28·7) | |
| | | | More – 4937/8109 (60·9) | |
| | | | Less – 843/8109 (10·4) | |
| | No | 5142/13251 (38·8) | Same – 1850/5142 (36·0) | |
| | | | More – 3205/5142 (62·3) | |
| | | | Less – 87/5142 (1·7) | |
| Low mood/depression (n/N, %) | | | | |

*(Continued)*

**Table 1.** (Continued)

|  |  | Pre-lockdown | During lockdown | p-value |
|---|---|---|---|---|
|  | Yes | 7123/13262 (53·7) | Same – 2042/7123 (28·7) |  |
|  |  |  | More – 4345/7123 (61·0) |  |
|  |  |  | Less – 736/7123 (10·3) |  |
|  | No | 6139/13262 (46·3) | Same – 2684/6139 (43·7) |  |
|  |  |  | More – 3362/6139 (54·8) |  |
|  |  |  | Less – 93/6139 (1·5) |  |
| Anxiety (n/N, %) |  |  |  |  |
|  | Yes | 6730/13240 (50·8) | Same – 1871/6730 (27·8) |  |
|  |  |  | More – 4142/6730 (61·6) |  |
|  |  |  | Less – 717/6730 (10·6) |  |
|  | No | 6510/13240 (49·2) | Same – 3171/6510 (48·7) |  |
|  |  |  | More – 3269/6510 (50·2) |  |
|  |  |  | Less – 70/6510 (1·1) |  |

Paired t-test for continuous variables, McNemar's test for categorical variables

SD: standard deviation

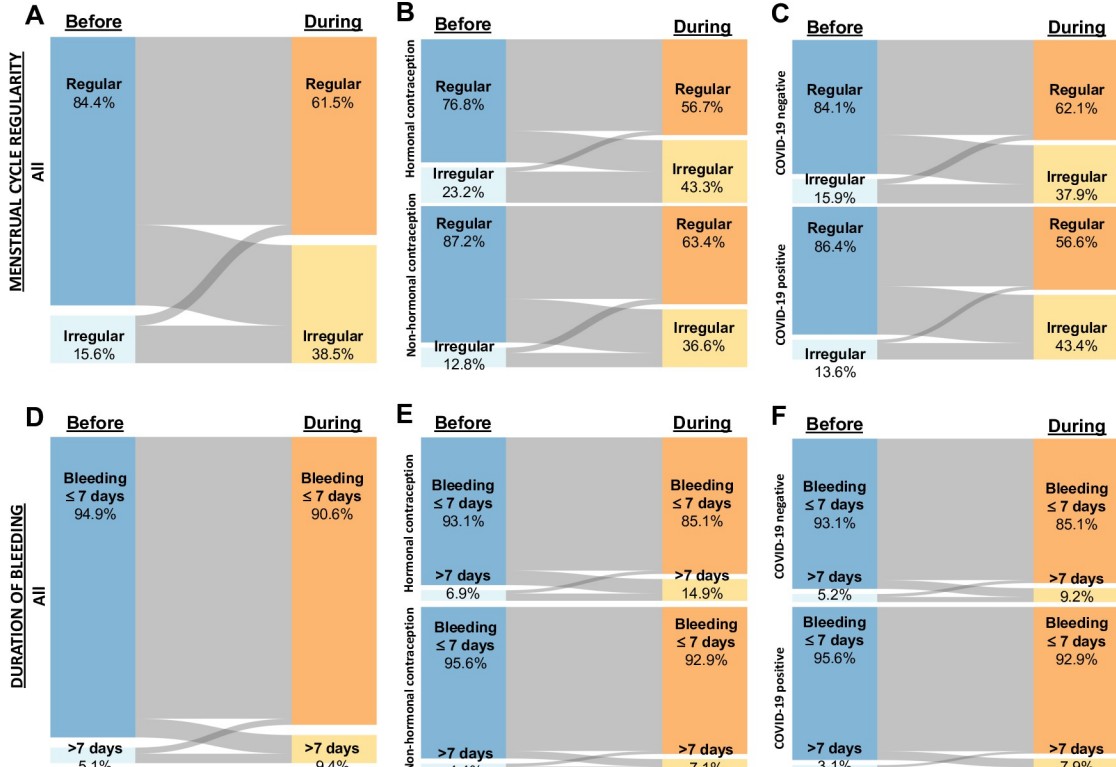

**Fig 1.** Menstrual cycle regularity and duration of bleeding in all participants (A & D), according to hormonal contraceptive use (B & E) and COVID-19 infection status (C & F). Nodes represent the proportion of participants in each category before (left) and during lockdown (right) with the flows between representing the proportional change between categories between the two timepoints.

**Table 2. Menstrual cycle characteristics, hormonal contraception vs no hormonal contraception.**

| | | Hormonal contraception user | | Non-hormonal contraception user | | p-value |
|---|---|---|---|---|---|---|
| Age, (mean, SD) | | 27·9 (0·8) | | 30·9 (0·7) | | <0·001 |
| Ethnicity (n/N, %) | Caucasian | 4507/4893 (92·1) | | 9050/10421 (86·8) | | <0·001 |
| | Black | 24/4893 (0·5) | | 111/10421 (1·1) | | |
| | Asian | 205/4893 (4·2) | | 823/10421 (7·9) | | |
| | Mixed | 127/4893 (2·6) | | 327/10421 (3·1) | | |
| | Arab | 10/4893 (0·2) | | 48/10421 (0·5) | | |
| | Other | 20/4893 (0·4) | | 62/10421 (0·6) | | |
| Parity | Parous | 618/4894 (12·6) | | 2812/10427 (27·0) | | <0·001 |
| (n/N, %) | Nulliparous | 4276/4894 (87·4) | | 7615/10427 (73·0) | | |
| Regular menstrual cycle (n/N, %) | Pre-lockdown | 2602/3390 (76·8) | | 7861/9012 (87·2) | | <0·001 |
| | During lockdown | 2040/3596 (56·7) | | 5602/8834 (63·4) | | <0·001 |
| Bleeding duration >7days (n/N, %) | Pre-lockdown | 230/3315 (6·9) | | 393/8989 (4·4) | | <0·001 |
| | During lockdown | 529/3549 (14·9) | | 632/8849 (7·1) | | <0·001 |
| Duration of bleeding, days (mean, SD) | Pre-lockdown | 5·2 (2·6) | | 5·1 (1·7) | | 0·081 |
| | During lockdown | 5·7 (3·6) | | 5·2 (2·3) | | <0·001 |
| Menstrual cycle length, days (mean, SD) | Pre-lockdown | 28·2 (17·8) | | 30·1 (10·4) | | <0·001 |
| | During lockdown | 28·5 (17·6) | | 30·8 (12·1) | | <0·001 |
| Painful periods (n/N, %) | Pre-lockdown | Yes – 1479/3211 (46·1) | No – 1732/3211 (53·9) | Yes – 5431/8722 (62·3) | No – 3291/8722 (37·7) | |
| | During lockdown | Same – 729/1479 (49·3) | Same – 993/1732 (57·3) | Same – 2659/5431 (49·0) | Same – 1915/3291 (58·2) | |
| | | More – 506/1479 (34·2) | More – 720/1732 (41·6) | More – 1759/5431 (32·4) | More – 1328/3291 (40·3) | |
| | | Less – 244/1479 (16·5) | Less – 19/1732 (1·1) | Less – 1013/5431 (10·6) | Less – 48/3291 (1·5) | |
| Heavy periods (n/N, %) | Pre-lockdown | Yes – 994/3185 (31·2) | No – 2191/3185 (68·8) | Yes – 4779/8721 (54·8) | No – 3942/8721 (45·2) | |
| | During lockdown | Same – 396/994 (39·8) | Same – 1316/2191 (60·1) | Same – 2297/4779 (48·1) | Same – 2579/3942 (65·4) | |
| | | More – 202/994 (20·4) | More – 631/2191 (28·8) | More – 1054/4779 (22·0) | More – 1083/3942 (27·5) | |
| | | Less – 396/994 (39·8) | Less – 244/2191 (11·1) | Less – 1428/4779 (29·9) | Less – 280/3942 (7·1) | |
| Intermenstrual spotting/bleeding (n/N, %) | Pre-lockdown | Yes – 980/4168 (23·5) | No – 3188/4168 (76·5) | Yes – 1162/9013 (12·9) | No – 7851/9013 (87·1) | |
| | During lockdown | Same – 277/980 (28·3) | Same – 2187/3188 (68·6) | Same – 453/1162 (39·0) | Same – 6566/7851 (83·6) | |
| | | More – 317/980 (32·3) | More – 947/3188 (29·7) | More – 287/1162 (24·7) | More – 1204/7851 (15·3) | |
| | | Less – 386/980 (39·4) | Less – 54/3188 (1·7) | Less – 422/1162 (36·3) | Less – 81/7851 (1·1) | |
| Abdominal bloating (n/N, %) | Pre-lockdown | Yes – 2854/4208 (67·8) | No – 1354/4208 (32·2) | Yes – 6518/9092 (71·7) | No – 2574/9092 (28·3) | |
| | During lockdown | Same – 1540/2854 (54·0) | Same – 896/1354 (66·2) | Same – 3717/6518 (57·0) | Same – 1749/2574 (68·0) | |
| | | More – 1103/2854 (38·6) | More – 441/1354 (32·5) | More – 2254/6518 (34·6) | More – 801/2574 (31·1) | |
| | | Less – 211/2854 (7·4) | Less – 17/1354 (1·3) | Less – 547/6518 (8·4) | Less – 24/2574 (0·9) | |
| Irritability (n/N, %) | Pre-lockdown | Yes – 2555/4181 (61·1) | No – 1626/4181 (38·9) | Yes – 6425/9049 (71·0) | No – 2624/9409 (29·0) | |

*(Continued)*

**Table 2.** (Continued)

| | | Hormonal contraception user | | Non-hormonal contraception user | | p-value |
|---|---|---|---|---|---|---|
| | During lockdown | Same – 826/2555 (32·3) | Same – 722/1626 (44·4) | Same – 2196/6425 (34·2) | Same – 1205/2624 (45·9) | |
| | | More – 1536/2555 (60·1) | More – 889/1626 (54·7) | More – 3529/6425 (54·9) | More – 1382/2624 (52·7) | |
| | | Less – 193/2555 (7·6) | Less – 15/1626 (0·9) | Less – 700/6425 (10·9) | Less – 37/2624 (1·4) | |
| Fatigue (n/N, %) | Pre-lockdown | Yes – 2320/4207 (55·1) | No – 1887/4207 (44·9) | Yes – 5789/9044 (64·0) | No – 3255/9044 (36·0) | |
| | During lockdown | Same – 629/2320 (27·1) | Same – 690/1887 (36·6) | Same – 1700/5789 (29·4) | Same – 1160/3255 (35·6) | |
| | | More – 1482/2320 (63·9) | More – 1158/1887 (61·4) | More – 3455/5789 (59·7) | More – 2047/3255 (62·9) | |
| | | Less – 209/2320 (9·0) | Less – 39/1887 (2·0) | Less – 634/5789 (10·9) | Less – 48/3255 (1·4) | |
| Low mood/depression (n/N, %) | Pre-lockdown | Yes – 2013/4204 (47·9) | No – 2189/4202 (52·1) | Yes – 5110/9060 (56·4) | No – 3950/9060 (43·6) | |
| | During lockdown | Same – 557/2013 (27·7) | Same – 914/2189 (41·8) | Same – 1485/5110 (29·1) | Same – 1770/3950 (44·8) | |
| | | More – 1279/2013 (63·5) | More – 1242/2189 (56·7) | More – 3066/5110 (60·0) | More – 2120/3950 (53·7) | |
| | | Less – 177/2013 (8·8) | Less – 33/2189 (1·5) | Less – 559/5110 (10·9) | Less – 60/3950 (1·5) | |
| Anxiety (n/N, %) | Pre-lockdown | Yes – 2005/4204 (47·7) | No – 2199/4204 (52·3) | Yes – 4725/9036 (52·3) | No – 4311/9036 (47·7) | |
| | During lockdown | Same – 559/2005 (27·9) | Same – 1050/2199 (47·7) | Same – 1312/4725 (27·8) | Same – 2121/4311 (49·2) | |
| | | More – 1277/2005 (63·7) | More – 1132/2199 (51·5) | More – 2865/4725 (60·6) | More – 2137/4311 (49·6) | |
| | | Less – 169/2005 (8·4) | Less – 17/2199 (0·8) | Less – 548/4725 (11·6) | Less – 53/4311 (1·2) | |
| Change in libido (n/N, %) | During lockdown | Same – 1813/4875 (37·2) | | Same – 4516/10179 (44·4) | | |
| | | Increased – 906/4875 (18·6) | | Increased – 2125/10179 (20·9) | | |
| | | Decreased – 2156/4875 (44·2) | | Decreased – 3538/10179 (34·8) | | |

Unpaired t-test for continuous variables, Chi$^2$ test for categorical variables

SD: standard deviation

irregular cycles (0·74 (0·66–0·83)) and bleeding duration more than 7 days (0·68 (0·59–0·78)) (**Table 2**). A significant increase in the duration of bleeding (pre pandemic 5·1 days vs during 5·3 days; p<0·001) and length of the menstrual cycle during the lockdown was also noted (pre pandemic 29·6 days vs during 30·1 days; p<0·001). **Fig 1** demonstrates menstrual cycle regularity and duration of bleeding in all participants (A, & D), according to hormonal contraceptive use (B & E) and COVID-19 infection status (C & F).

Dysmenorrhoea was the most prevalent bleeding-related symptom experienced by 57·9% (n = 6,910) of respondents prior to lockdown and 32·8% (n = 2,265) reported worsening pain, whilst 18·2% (n = 1,257) reported an improvement. Approximately 40·8% (n = 2,048) of those who did not experience dysmenorrhoea prior to lockdown reported an increased prevalence (**Table 1** and **Fig 2**).

Heavy periods were reported prior to the lockdown by 48·5% (n = 5,773) of participants, whereby the majority reported no change during lockdown (n = 2,693; 46·6%) (**Fig 2B**). However, 31·6% (n = 1,824) with heavy periods experienced a decrease in bleeding and 21·8% (n = 1,256) reported an increase (**Fig 2B**). Almost one-third of those who did not previously experience heavy periods reported an increase during the lockdown (n = 1,714; 28·0%) (**Fig 2B**). IMB was reported by 16·3% (n = 2,142) prior to lockdown, and in this group 28·2%

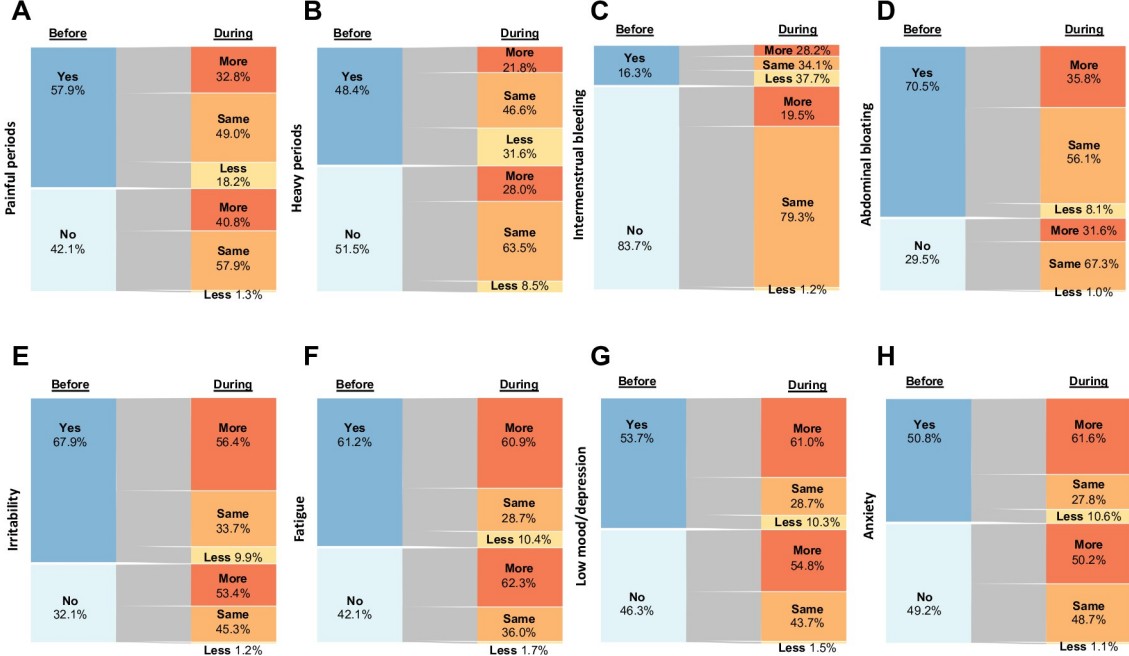

**Fig 2. Changes in menstrual cycle characteristics and premenstrual symptoms in all participants.** Nodes represent the proportion of participants reporting each symptom before lockdown (left) and the proportion reporting whether they experienced this symptom more, the same or less during lockdown (right).

(n = 604) noticed an increase. Of the 83·7% (n = 11,039) who did not experience IMB previously, 19·5% (n = 2.151) began to experience it during the lockdown (**Fig 2C**).

Participants were asked about ten PMS, taken from the 'Daily Record of Severity of Problems' [17]. Abdominal bloating, irritability and fatigue were the most frequently reported symptoms prior to lockdown (n = 9,372; 70·5%, n = 8,980; 67·9% and n = 8,109; 61·2% respectively) (**Fig 2D–2F**). Premenstrual low mood/depression, anxiety and irritability were most frequently noted to increase during lockdown, with the majority of participants reporting an increase, irrespective of whether they experienced it before the lockdown (**Table 2**).

## Hormonal contraception

Half of the participants were using some form of contraception (50·2%), and 31.4% were using a hormonal method (n = 4896). The combined oral contraceptive pill (COCP) was the most popular method (n = 2,328; 27·8% of all contraceptive users), followed by condoms (n = 2,027; 24·2%), the progestogen intra-uterine system (IUS) (n = 1,090; 13·0%) and the copper intra-uterine device (IUD) (n = 846; 10·1%). Eight percent (n = 600) of contraceptive users reported difficulty in accessing contraception and this was most commonly reported by those using the COCP.

**Table 2** **and Fig 1** represents the dataset stratified according to use of hormonal versus non-hormonal contraception. Hormonal contraception [HC] users were more likely to be younger, Caucasian and nulliparous compared to non-users (all p<0·001). Whilst HC users were more likely to report irregular menstrual cycles both before and during the lockdown (p<0·001), HC use was associated with a decreased risk of developing irregular cycles (0·57, (0·52–0·61)) (**Table 2**), with non-HC users reporting a 3-fold increase in cycle irregularity, compared to a 2-fold increase in HC users (**Table 2 and Fig 1**). The mean duration of bleeding was similar between the two groups prior to lockdown (HC 5·2 vs. non-HC user 5·1 days;

p = 0·081). HC users experienced the greatest increase during lockdown resulting in this group having a greater bleeding duration than non-users (5·7 vs 5·2 days; p<0·001). Bleeding duration greater than seven days was reported by twice the number of HC users compared to non-users during lockdown (14·9% vs 7·1%; p<0·001). The overall duration of the menstrual cycle was longer in non-users both before (HC 28·2 vs non-HC user 30·1 days; p<0·001) and during the lockdown (HC 28·5 vs non-HC user 30·8 days; p<0·001).

Heavy and painful periods were more frequently experienced by non-HC-users before the lockdown, and both groups reported similar trends in these factors increasing, decreasing, or remaining the same during the lockdown. PMS symptoms were slightly more common amongst non-HC-users prior to lockdown, and again similar trends relating to the changes observed during the lockdown were seen in both HC and non-HC users. A greater proportion of HC users reported a decrease in libido during the lockdown (HC 44·2% vs non-HC user 34·8%) (Table 2).

## COVID-19 infection

Overall, 2·4% (n = 371) of participants reported a history of COVID-19 infection, confirmed by either polymerase chain reaction (PCR) swab or antibody test. A further 14·2% (n = 2210) of all respondents reported symptoms of COVID-19 but were not tested (Table 1).

Table 3 represents a sub-group analysis of the confirmed COVID-19 infections versus negative participants, excluding those with a suspected infection. No difference was seen in age or parity between the two groups. However, participants reporting a COVID-19 infection were less likely to be Caucasian (COVID-19 positive 83·7% non-Caucasian vs negative 88·8% Caucasian; p<0·001) and more likely to use HC (HC 42·1% vs non-HC user 32·4%; p = 0·012). No significant difference was found in cycle regularity, bleeding duration greater than seven days, or duration of bleeding according to COVID-19 infection (Table 1). The overall length of the menstrual cycle was shorter prior to lockdown in those who had a COVID-19 infection (COVID-19 positive 27·9 vs negative 29·7 days; p = 0·019). However, no difference during the lockdown (COVID-19 positive 29·3 vs negative 30·2 days; p = 0·264) was seen. Similar trends in rates of PMS symptoms both before and during the lockdown were observed in both groups. Changes in libido were also comparable (Table 3).

## Lifestyle during COVID-19 lockdown

Table 3 reports the lifestyle changes during the lockdown. Most reported commuting less (n = 8457; 81·1%) and 24·2% (n = 2211) were home-schooling. With regards to exercise, 42·5% (n = 5661) exercised more, 38·7% (n = 5149) less and 18·8% (n = 2508) carried out the same amount of exercise during the pandemic. A large proportion of the sampled cohort reported an increase in the amount of yoga and pilates (n = 4645; 51·4%), followed by high-intensity interval training (n = 4078; 48·6%), cycling (n = 2903; 48·7%), running (n = 4396; 48·6%) and walking (n = 6475; 48·5%). Conversely, the majority participated in less weight-training (n = 4561; 56·5%).

Changes relating to sleep were common, with 53·1% (n = 7065) reporting worsened sleep quality and 51·8% (n = 6932) reporting greater difficulty falling asleep. Conversely, 14·9% (n = 1988) reported better quality and 14·5% (n = 1948) greater ease in falling asleep, with 26·4% (n = 3517) also reporting a more consistent sleep pattern. Changes in diet were also common with 30·2% (n = 4038) reporting what they considered to be a healthier diet, 34·0% (n = 4532) less healthy and 35·8% (n = 4783) the same. 36·5% (n = 4854) reported drinking more alcohol and 34·1% (n = 4520) drank less.

**Table 3. Menstrual cycle characteristics, Covid-19 positive (confirmed only) vs Covid-19 negative.**

| | | COVID-19 positive (confirmed only) | | COVID-19 negative | | p-value |
|---|---|---|---|---|---|---|
| Age, (mean, SD) | | 29·7 (6·3) | | 29·9 (6·5) | | 0·590 |
| Ethnicity (n/N, %) | Caucasian | 309/369 (83·7) | | 11188/12600 (88·8) | | <0·001 |
| | Black | 8/369 (2·2) | | 102/12600 (0·8) | | |
| | Asian | 42/369 (11·4) | | 840/12600 (6·7) | | |
| | Mixed | 7/369 (1·9) | | 359/12600 (2·9) | | |
| | Arab | 2/369 (0·5) | | 48/12600 (0·4) | | |
| | Other | 1/369 (0·3) | | 63/12600 (0·5) | | |
| Parity (n/N, %) | Parous | 636/2579 (24·7) | | 2745/12601 (21·8) | | 0·086 |
| | Nulliparous | 1943/2579 (75·3) | | 9856/12601 (78·2) | | |
| Contraception type (n/N, %) | Hormonal | 156/371 (42·1) | | 4086/12613 (32·4) | | 0·012 |
| | Non-hormonal | 215/371 (57·9) | | 8527/12613 (67·6) | | |
| Bleeding duration >7days (n/N, %) | Pre-lockdown | 9/287 (3·1) | | 531/10168 (5·2) | | 0·115 |
| | During lockdown | 23/290 (7·9) | | 944/10252 (9·2) | | 0·458 |
| Regular menstrual cycle, (n/N, %) | Pre-lockdown | 253/330 (86·4) | | 8619/10250 (84·1) | | 0·296 |
| | During lockdown | 167/295 (56·6) | | 6386/10282 (62·1) | | 0·055 |
| Menstrual cycle length, days (mean, SD) | Pre-lockdown | 27·9 (14·0) | | 29·7 (13·1) | | 0·019 |
| | During lockdown | 29·3 (15·5) | | 30·2 (12·9) | | 0·264 |
| Duration of bleeding, days (mean, SD) | Pre-lockdown | 5·0 (1·9) | | 5·1 (2·0) | | 0·358 |
| | During lockdown | 5·2 (2·4) | | 5·3 (2·7) | | |
| Painful periods (n/N, %) | Pre-lockdown | 160/281 (56·9) | 121/281 (43·1) | 5682/9868 (57·6) | 4186/9869 (42·4) | |
| | During lockdown | Same – 80/160 (50·0) | Same – 68/121 (56·2) | Same – 2869/5682 (50·5) | Same – 2465/4186 (58·9) | |
| | | More – 39/160 (30·6) | More – 52/121 (43·0) | More – 1818/5682 (32·0) | More – 1667/4186 (39·8) | |
| | | Less – 31/160 (19·4) | Less – 1/121 (0·8) | Less – 995/5682 (17·5) | Less – 54/4186 (1·3) | |
| Heavy periods (n/N, %) | Pre-lockdown | 114/275 (41·5) | 161/275 (58·5) | 4745/9850 (48·2) | 5105/9850 (51·8) | |
| | During lockdown | Same – 57/114 (50·0) | Same – 89/161 (55·3) | Same – 2264/4745 (47·7) | Same – 3311/5105 (64·9) | |
| | | More – 21/114 (18·4) | More – 55/161 (34·2) | More – 1014/4745 (21·4) | More – 1371/5105 (26·9) | |
| | | Less – 36/114 (31·6) | Less – 17/161 (10·5) | Less – 1467/4745 (30·9) | Less – 423/5105 (8·2) | |
| Intermenstrual spotting/bleeding (n/N, %) | Pre-lockdown | 60/334 (18·0) | 274/334 (82·0) | 1757/10896 (16·1) | 9139/10896 (83·9) | |
| | During lockdown | Same – 19/60 (31·7) | Same – 211/274 (77·0) | Same – 595/1757 (33·9) | Same – 7284/9139 (79·7) | |
| | | More – 20/60 (33·3) | More – 62/274 (22·6) | More – 496/1757 (28·2) | More – 1748/9139 (19·1) | |
| | | Less – 21/60 (35·0) | Less – 1/274 (0·4) | Less – 666/1757 (37·9) | Less – 107/9139 (1·2) | |
| Abdominal bloating (n/N, %) | Pre-lockdown | 238/331 (71·9) | 93/331 (28·1) | 7694/10998 (70·0) | 3304/10998 (30·0) | |
| | During lockdown | Same – 135/238 (56·7) | Same – 58/93 (62·4) | Same – 4380/7694 (56·9) | Same – 2261/3304 (68·4) | |
| | | More – 93/238 (39·1) | More – 32/93 (34·4) | More – 2695/7694 (35·0) | More – 1013/3304 (30·7) | |
| | | Less – 10/238 (4·2) | Less – 3/93 (3·2) | Less – 619/7694 (8·1) | Less – 30/3304 (0·9) | |
| Irritability (n/N, %) | Pre-lockdown | 217/327 (66·4) | 110/327 (33·6) | 7379/10941 (67·4) | 3562/10941 (32·6) | |

*(Continued)*

**Table 3.** (Continued)

| | | COVID-19 positive (confirmed only) | | COVID-19 negative | | p-value |
|---|---|---|---|---|---|---|
| | During lockdown | Same – 88/217 (40·6) | Same – 52/110 (47·3) | Same – 2517/7379 (34·1) | Same – 1649/3562 (46·3) | |
| | | More – 114/217 (52·5) | More – 55/110 (50·0) | More – 4116/7379 (55·8) | More – 1875/3562 (52·6) | |
| | | Less – 15/217 (6·9) | Less – 3/110 (2·7) | Less – 746/7379 (10·1) | Less – 38/3562 (1·1) | |
| Fatigue (n/N, %) | Pre-lockdown | 186/332 (56·0) | 146/332 (44·0) | 6713/10963 (61·2) | 4250/10963 (38·8) | |
| | During lockdown | Same – 39/186 (21·0) | Same – 48/146 (32·9) | Same – 1996/6713 (29·7) | Same – 1594/4250 (37·5) | |
| | | More – 139/186 (74·7) | More – 95/146 (65·1) | More – 3999/6713 (59·6) | More – 2686/4250 (60·9) | |
| | | Less – 8/186 (4·3) | Less – 3/146 (2·0) | Less – 718/6713 (10·7) | Less – 70/4250 (1·6) | |
| Low mood/depression (n/N, %) | Pre-lockdown | 154/331 (46·5) | 177/331 (53·5) | 5840/10975 (53·2) | 5135/10975 (46·8) | |
| | During lockdown | Same – 65/154 (42·2) | Same – 93/177 (52·5) | Same – 1679/5840 (28·8) | Same – 2281/5135 (44·4) | |
| | | More – 81/154 (52·6) | More – 83/177 (46·9) | More – 3548/5840 (60·7) | More – 2272/5135 (54·0) | |
| | | Less – 8/154 (5·2) | Less – 1/177 (0·6) | Less – 613/5840 (10·5) | Less – 82/5135 (1·6) | |
| Anxiety (n/N, %) | Pre-lockdown | 147/333 (44·1) | 186/333 (55·9) | 5501/10952 (50·2) | 5451/10952 (49·8) | |
| | During lockdown | Same – 46/147 (31·3) | Same – 111/186 (59·7) | Same – 1543/5501 (28·1) | Same – 2670/5451 (49·0) | |
| | | More – 91/147 (61·9) | More – 74/186 (39·8) | More – 3365/5501 (61·2) | More – 2719/5451 (49·9) | |
| | | Less – 10/147 (6·8) | Less – 1/186 (0·5) | Less – 593/5501 (10·8) | Less – 62/5451 (1·1) | |
| Change in libido (n/N, %) | During | Same – 154/371 (41·5) | | Same – 5318/12486 (42·6) | | |
| | lockdown | Increased – 66/371 (17·8) | | Increased – 2532/12486 (20·2) | | |
| | | Decreased – 151/371 (40·7) | | Decreased – 4636/12486 (37·1) | | |

Unpaired t-test for continuous variables, $Chi^2$ test for categorical variables

SD: standard deviation

The most frequently reported source of anxiety during the lockdown was the health of others (n = 11842; 89·0%), followed by the news (n = 11432; 86·2%), anxiety for which they could not determine a cause (n = 10137; 77·1%), their own health (n = 8310; 62·7%), feeling lonely (n = 7521; 57·2%) and work (n = 6671; 53·7%).

## Discussion

This is the largest study to date documenting the impact of the COVID-19 pandemic on menstrual cycles, hormonal symptoms and lifestyle changes, with over 15,500 participants. Using an online survey, we have demonstrated a full spectrum of changes to the menstrual cycle including regularity, dysmenorrhoea, volume of bleeding and IMB, with three-quarters of the participants reporting a change in at least one of these factors.

There was a significant increase in irregular menstrual cycles reported during the pandemic, compared to prior. This is consistent with a study which reported an incidence of 27.6% during the pandemic, compared to 12.1% before (P = 0.008) [20]. Psychological stress is a risk factor for hypothalamic hypogonadism causing infrequent or absent periods [21]. This may explain why cycle irregularity was particularly prevalent amongst women who started working from home during the pandemic, with many finding this a stressful transition [22]. Understanding of cycle regularity during times of stress is important, because irregularity of

periods is also associated with pregnancy risks, including pre-eclampsia, low birth weight, spontaneous delivery and increased incidence of metabolic disorders [23, 24]. Thus, the impact of the pandemic may have long term consequences spanning across women's health into antenatal care, further burdening the demand on healthcare services.

We also observed a statistically significant increase in the length of the menstrual cycle and duration of bleeding. However, these were 0.4 days and 0.2 days respectively, which are of limited clinical value. Further studies however, have since demonstrated inconsistent findings. In a study amongst women aged 18–45, there was a decrease in the number of pads used during periods amidst the pandemic (3.7 ± 2.6 pads/day before pandemic vs. 3.2 ± 1.5 pads/day during pandemic) [25]. Duration of the cycle was also decreased (6.3 ± 2.1 days prior pandemic vs. 5.9 ± 1.8 days during pandemic) [25] and the length of cycles decreased from 29.40 days before the pandemic, to 29.12 days during (P< 0.001) [26]. Conversely however, in a study of 269 participants, 44.4% of women observed an increase in duration of their cycle during the pandemic [27]. Notably, the incidence was higher in women who were affected by the pandemic or had a family member who was, compared to women who were not affected (53.9% vs. 46.1%) [27]. This further highlights the relevance of COVID-19 related stress on menstrual cycles.

A large proportion of women also reported heavier and/or more painful periods, which is consistent with a recent systematic review which identified prolonged duration of the cycle and increased episodes of pain, as the main changes to cycles during the pandemic [26, 28, 29]. Furthermore, in a study of 1,031 participants, amongst the 47% of women who experienced heavy periods, 5% reported an increased incidence during the pandemic (P = 0.003) and amongst 49% who reported painful periods, 7% stated that this symptom increased (P< 0.0001) [30]. Moreover, women who experienced mental health symptoms were more likely to report painful periods [30]. This has also been demonstrated in a study whereby an increased severity of dysmenorrhoea was reported amongst women experiencing COVID-19 related anxiety and depression (P = 0.025 vs. P = 0.008 respectively) [31]. The evolvement of non-painful periods to painful was also associated with COVID-19 related anxiety [32]. It is therefore likely that menstrual related symptoms are exacerbated by environmental changes and stress encountered by pandemic related issues, such as privacy, access to and affordability of menstrual products and reduced availability and accessibility of sexual and reproductive health care services [33]. Specific stressors related to the pandemic have also included work related stress, change in financial situation, difficulties with home schooling or childcare, family or partner conflict or family illness or bereavement [30].

A high number of the cohort reported worsening PMS, most notably low mood/depression, anxiety and irritability. This is consistent with a survey of 385 medical students, where an increase in PMS symptoms such as emotional disturbance, weakness, mastalgia and sleep disturbance during the pandemic was observed [31]. In addition, a study of 400 college students identified a 19% increase in the prevalence of premenstrual dysphoric disorder and 43.3% uptake of premenstrual syndrome amongst the cohort during the pandemic [34]. This is perhaps unsurprising, given that PMS symptoms are often related to stressful events [35, 36].

Evidence suggests that women who suffer from PMS exhibit a higher incidence of psychological disorders such as depression, anxiety and stress [37–39]. Stress can modulate and inhibit menstrual cycle function via a sophisticated crosstalk between the hypothalamic-pituitary-gonadal (HPG) and hypothalamic-pituitary-adrenal (HPA) axes. This process interferes with gonadotrophin secretion, luteinizing hormone release, oestrogen and progesterone production and can result in anovulation [40]. The fluctuation of reproductive hormones during the cycle results in premenstrual symptoms [41]. This can manifest as irregular cycles with IMB as well as prolonged, heavy and painful periods [42].

Whilst we did not screen specifically for anxiety or depression, a general deterioration in mental health has been observed throughout the COVID-19 pandemic, with a large UK study of 53, 351 individuals demonstrating the greatest decline in women, young people and those with young children [43].

Various studies have also demonstrated that women of reproductive age are more susceptible to symptoms of depression and anxiety during the pandemic, compared to men [44–46]. Given that a large proportion of our cohort reported worry and anxiety related to the pandemic and its impact on their health, families, careers and finance and an increase in PMS, may suggest they are at risk of developing mental health illnesses. This may be attributed to the COVD-19 mitigation and control strategies, such as lockdowns and social distancing leading to increased psychological stress, potentially resulting in disturbances to the HPG axis and neuronal circuits for many women, worsening the severity of PMS symptoms overall [41]. An increase in mental health illnesses amongst women will undoubtedly burden the national health service financially, as illnesses then persist through pregnancy and the post-partum period, causing significant maternal morbidity. As such, appropriate interventions should be adopted within public health strategies [47], to support women experiencing hormonal changes during the pandemic. An example of this includes the successful implementation of a number of online psychological self-help resources, such as cognitive behavioural therapy for depression, anxiety and insomnia and several programmes through artificial intelligence (AI) which have been developed to aid people experiencing psychological distress during the pandemic in China [48].

A small proportion of our cohort reported a confirmed COVID-19 infection prior to completing the survey. We were able to deduce there was no significant difference between women who contracted the virus and those who did not, in regards to the impact on menstrual cycle characteristics. This further supports the notion that changes observed overall, are likely to result from the demonstrable change in lifestyle or the psychological impact, rather than contraction of the virus. A previously published study did not reveal any significant difference in menstrual cycle characteristics or sex hormone concentrations between 147 women with mild, confirmed COVID-19 infection and 90 with a severe COVID-19 infection in China [49]. However, the study mentioned did not include a control group of uninfected individuals.

Notably, almost half of the participants reported increased exercise levels. Whilst excessive exercise can cause irregular or absent periods [50], regular moderate exercise can be beneficial for painful periods [51] and PMS symptoms [52]. This could explain why some members of the study cohort reported an improvement in these factors. Around half the participants reported a reduction in sleep quality, quantity and difficulty falling asleep, which appears highly prevalent during the COVID-19 pandemic [53]. This supports a study which showed that women who experienced menstrual irregularity during the pandemic experienced significant changes and fluctuations in their bedtime (P<0.01) [28]. This is important considering poor sleep is a significant predictor of missed periods and menstrual cycle disruption [32]. Caffeine and alcohol, both of which many participants reported an increased intake, can also interfere with sleep, with the latter associated with a higher risk of PMS symptoms on dose-dependent scale [54, 55]. Many of these factors may be interrelated, which highlights the difficulty in assessing cause and effect with regards to lifestyle and hormonal symptoms. Evidently, further understanding of the influence of environmental factors on the menstrual cycle is still required.

The adverse changes in menstrual cycle characteristics and hormonal symptoms experienced during the pandemic is concerning for several reasons. Women are disproportionately affected by the pandemic, as they bare the majority of childcare responsibilities, which consequently predisposes them to insecure employment and finances [15]. Prior to the pandemic, it

was well established that dysmenorrhoea negatively impacts family and social relationships, and considered problematic with regards to quality of life and limiting daily activities [56]. For example, menstrual cycle dysfunction results in absenteeism, and it is estimated that women with heavy periods work on average 3.6 weeks less per year than those without [57]. Whilst the act of simply attending work may not be required for the large numbers of women working from home during the pandemic, that does not mean they retain the ability to carry out the tasks required of them, and in fact presenteeism, that is productivity loss whilst attending work, may present an equally important economic burden [58]. We must therefore take into account the healthcare spend to evaluate and where necessary treat these women [59].

Menstrual cycle changes may also have detrimental effects on society. In particular, the potential to negatively affect fertility rates. Ovulation is required for a pregnancy, and not only does anovulation present an obvious problem for those wishing to conceive, but stress, as measured by saliva alpha-amylase levels, has been associated with a reduction in fecundability [60], greater time to pregnancy and risk of infertility [61]. Given that a large proportion of participants reported a decrease in libido, we recommend that further studies should also consider the relationship between the pandemic and consequence on fertility rates.

Our findings are particularly important when counselling women of reproductive age, who may still exhibit feelings of vaccine hesitancy, and therefore reluctant to be vaccinated because of pregnancy related fears. Reassuringly, one of the largest studies of 26,710 from the UK, reported that 80% of pre-menopausal vaccinated women (n = 4,989) did not experience menstrual cycle changes up to 4 months following the first COVID-19 injection [62]. Furthermore, a systematic review which did demonstrate menstrual abnormalities following COVID-19 vaccination, identified that the changes were intermittent and self-limiting [3]. Despite our findings that the pandemic did result in menstrual cycle changes, it is apparent this was primarily associated with specific pandemic related stressors, rather than from the virus itself. Therefore, if women were able to effectively manage these life stressors, they may be able to limit the impact on their menstrual cycles. Furthermore, it is important for clinicians to continue to explore vaccination concerns, engage in discussions regarding risks and benefits of vaccination prior to pregnancy and educate women about current evidence, particularly amongst vulnerable black, Asian and minority ethnic groups, so that they can make well informed decisions regarding their reproductive health.

Our study does of course have its limitations. First, individuals who noticed a change in their menstrual cycle may feel more motivated to complete the questionnaire, which may represent the significant change reported. Second, retrospective self-reporting of menstrual cycle characteristics may be subject to inherent inaccuracy and recall bias [63]. However, we note that almost 70% of respondents formally tracked their menstrual cycle using either an app, diary or other tracking method which provides additional reassurance of the reliability of the data provided. Third, the process of snowballing recruitment is limited because re-sharing of the study information can substantially increase the number of potential responses that can be obtained. The disadvantage being that recruitment remains biased towards social media users, who may not be representative of the general population, thus introducing a degree of selection bias. Fourth, when reporting on data from such a large, heterogenous population, there will be unknown confounders that cannot be taken into account, which may under- or over represent the associations that we report in this study. For this reason we decided against attempting to correlate menstrual cycle changes in women who had experienced COVID-19 infection prior to completing the study, to prevent potential overinterpretation of the data. Particularly, with such a heterogenous population, the result is likely to be highly individualised, because of the differing baseline of that individual, degree of change and the complex interplay of these specific factors. We also cannot overlook the large number of participants

who reported an improvement in pain, bleeding, and PMS, and therefore cannot simply conclude that the pandemic has had a negative effect. Fifth, the low number of confirmed COVID cases may be explained by the low level of testing of the general population in the UK at the time of the questionnaire. We therefore decided to remove those who reported symptoms, but were not tested from this sub-group analysis to control for this issue. Finally, follow-up of this cohort was not conducted, and therefore it is unknown how long menstrual cycle changes persist. This would now be difficult to determine, because of subsequent emerging reports of further menstrual cycle disturbance following receipt of COVID-19 vaccines. Our study was conducted prior to any COVID-19 vaccination trials or programme roll-out, and therefore the potential impact of any vaccine is not reflected in our data. In addition, further lockdowns introduced by the UK government since the first one may have influenced interpretation of results. Future studies should consider the long term implications of menstrual cycle changes following the COVID-19 pandemic, as well as providing further education and support for women who are vulnerable to life stressors that may compromise their reproductive health.

## Conclusions

This is the largest study of menstrual cycle characteristics during the COVID-19 pandemic and has demonstrated that a significant number of women have experienced irregular menstrual cycles and prolonged episodes of bleeding, as well as changes in the amount of pain and bleeding, following the COVID-19 lockdown. Our data suggests this is primarily related to behavioural changes experienced during the pandemic, rather than from contraction of the virus itself.

Although menstrual cycle irregularities may initially seem a trivial consequence of a devastating pandemic, this can have a substantial impact on the individual, the economy, healthcare and on population levels. The wealth of a nation is heavily reliant on the health of its women which is the reason improving health and reducing gender inequality are highlighted as key areas for improvement in the United Nations Secretary General's 2030 Agenda for Sustainable Development [64]. We therefore recommend that this data should be taken into consideration by governments, healthcare providers and employers when developing further pandemic recovery plans and to work towards reducing health inequity.

## Supporting information

**S1 Checklist. STROBE statement—checklist of items that should be included in reports of observational studies.**
(DOCX)

**S1 Data.**
(CSV)

## Author Contributions

**Conceptualization:** Anita Mitra, Maria Kyrgiou, Srdjan Saso.

**Data curation:** Anita Mitra, Jan Y. Verbakel.

**Formal analysis:** Anita Mitra, Jan Y. Verbakel.

**Investigation:** Anita Mitra.

**Methodology:** Anita Mitra.

**Supervision:** Srdjan Saso.

**Writing – original draft:** Anita Mitra, Lorraine S. Kasaven.

**Writing – review & editing:** Lorraine S. Kasaven, Menelaos Tzafetas, Karen Grewal, Benjamin Jones, Phillip R. Bennett, Maria Kyrgiou, Srdjan Saso.

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
