## [Decision Letter · Decision Letter 0]

14 Jun 2023

PONE-D-23-13834The Menstrual Cycle and the COVID-19 PandemicPLOS ONE

Dear Dr. Kasaven,

Thank you for submitting your manuscript to PLOS ONE. After careful consideration, we feel that it has merit but does not fully meet PLOS ONE’s publication criteria as it currently stands. Therefore, we invite you to submit a revised version of the manuscript that addresses the points raised during the review process.

We look forward to receiving your revised manuscript.

Kind regards,

Muhammad Junaid Farrukh

Academic Editor

PLOS ONE

Reviewers' comments:

Reviewer's Responses to Questions

**Comments to the Author**

1. Is the manuscript technically sound, and do the data support the conclusions?

Reviewer #1: Yes

Reviewer #2: Yes

2. Has the statistical analysis been performed appropriately and rigorously? 

Reviewer #1: Yes

Reviewer #2: Yes

3. Have the authors made all data underlying the findings in their manuscript fully available?

Reviewer #1: Yes

Reviewer #2: Yes

4. Is the manuscript presented in an intelligible fashion and written in standard English?

Reviewer #1: Yes

Reviewer #2: Yes

5. Review Comments to the Author

Reviewer #1: The article is really good., I just have few points to share:

I was wondering how did the researchers select the social media platforms? why did they choose Instagram & Twitter?

Moreover, sample size calculation was not mentioned in the article

Reviewer #2: Title: A more specific or descriptive subtitle could potentially provide additional information about the focus or scope of the research.

The introduction does not explicitly highlight any novel insights. While the introduction mentions the need to understand the impact of the virus on unvaccinated women, it could further emphasize the importance of studying this population to provide comprehensive insights into the effects of COVID-19 on menstrual cycles.

To enhance the novelty of the study, you can highlight any unique aspects or gaps in the existing literature that the investigation aims to address.

Consider including a sentence or two about the potential implications or benefits of studying menstrual cycle changes in relation to COVID-19. This could further justify the importance and relevance of the research.

Method

Consider providing a brief explanation of why an online cross-sectional survey was chosen as the study design. This can help justify the choice and highlight the advantages or limitations of this approach.

Specify the target population more clearly. For example, mention that the survey targeted women of reproductive age or specify the age range explicitly.

The section could provide more detail on the survey questions and response options. This would help readers understand the specific menstrual cycle characteristics, hormonal symptoms, and lifestyle changes that were assessed in the study.

The section could provide more information on the sample size and characteristics of the study population. This would help readers understand the generalizability of the study findings and the representativeness of the sample.

Provide more details about the recruitment process. How were the social media advertisements designed? How were they targeted to reach the desired population? Including these details can give readers a better understanding of how participants were selected.

Consider mentioning the estimated sample size or the number of participants who took part in the study. This can help readers assess the representativeness and statistical power of the findings.

Discussion

There are a few areas where the discussion could be improved:

1. The section on the impact of the pandemic on menstrual cycles could be more concise. While the study's findings on menstrual cycle changes are important, the discussion could be streamlined to focus on the most significant findings and their implications.

2. The section on the impact of the pandemic on mental health could be expanded. The study's findings on the worsening of PMS symptoms and the high prevalence of anxiety and depression among participants are important, and the discussion could be expanded to explore the potential reasons for these findings and their implications for women's health.

6. PLOS authors have the option to publish the peer review history of their article (what does this mean?). If published, this will include your full peer review and any attached files.

Reviewer #1: **Yes: **Noor Al-Tameemi

Reviewer #2: No

While revising your submission, please upload your figure files to the Preflight Analysis and Conversion Engine (PACE) digital diagnostic tool, https://pacev2.apexcovantage.com/. PACE helps ensure that figures meet PLOS requirements. To use PACE, you must first register as a user. Registration is free. Then, login and navigate to the UPLOAD tab, where you will find detailed instructions on how to use the tool. If you encounter any issues or have any questions when using PACE, please email PLOS at figures@plos.org. Please note that Supporting Information files do not need this step.<quillbot-extension-portal></quillbot-extension-portal>

---

## [Author Response · Author response to Decision Letter 0]

4 Aug 2023

Miss Lorraine Kasaven MBChB BSc MRCOG

Clinical Research Fellow

Division of Surgery and Cancer

Imperial College London

Hammersmith Hospital Campus, 

Du Cane Road, London, W12 0NN

4th August 2023

Editorial Office 

PLOS ONE

Dear Editor in Chief,

Title: PONE-D-23-13834: ‘The Menstrual Cycle and the COVID-19 Pandemic’

We would like to take this opportunity to thank you and the reviewers for their expert comments. We have carefully read the feedback and have addressed the points accordingly, which we are certain has led to an overall improvement in the quality of the manuscript. A response to each of the reviewers points is outlined below. Alterations are shown in the revised manuscript using tracked changes as requested. 

Reviewer 1:

1)‘The article is really good. I just have few points to share: I was wondering how did the researchers select the social media platforms? why did they choose Instagram & Twitter? Moreover, sample size calculation was not mentioned in the article.’

We thank reviewer one for their feedback. We have now included further details regarding why the social media platforms ‘Instagram’ and ‘Twitter’ were used. We have also referred to the practice error margin used in social studies to determine what the optimal sample size from a survey should be. (Pg 5-6; Lines 138-142 and 172-176)

Reviewer 2: 

1)‘A more specific or descriptive subtitle could potentially provide additional information about the focus or scope of the research.’

We thank the reviewer for this comment. We have now changed the running title to ‘Did the COVID-19 pandemic cause changes to unvaccinated women’s menstrual cycles and hormonal symptoms?’ If the reviewer feels this would be an overall more appropriate title for the paper, we would be happy to change it. 

2) ‘The introduction does not explicitly highlight any novel insights. While the introduction mentions the need to understand the impact of the virus on unvaccinated women, it could further emphasize the importance of studying this population to provide comprehensive insights into the effects of COVID-19 on menstrual cycles.’

3) ‘To enhance the novelty of the study, you can highlight any unique aspects or gaps in the existing literature that the investigation aims to address.’

4) ‘Consider including a sentence or two about the potential implications or benefits of studying menstrual cycle changes in relation to COVID-19. This could further justify the importance and relevance of the research.’

We have now amended the introduction based on feedback points 2-4. We have further emphasised the importance of studying this population of women, with suggestions of what the comprehensive insights into the findings from the study will be. This includes the following 3 points:

1) Most information reported during the pandemic was through social media sites with users self-reporting their symptoms as opposed to reporting scientific evidence. Such misinformation may be fuelling vaccine hesitancy.

2) The scientific information obtained from this study can be used to guide economic recovery plans following the pandemic, by understanding how women’s menstrual cycles have been affected, preventing them from contributing to the workforce. 

3) The findings will be used to address gender bias when it comes to research and development, which is often neglected with respect to women’s health.

5) ‘Consider providing a brief explanation of why an online cross-sectional survey was chosen as the study design. This can help justify the choice and highlight the advantages or limitations of this approach.’

We have now explained the indications for using a cross sectional survey as the main study design. (Pg 5; Lines 133-136)

6) Specify the target population more clearly. For example, mention that the survey targeted women of reproductive age or specify the age range explicitly.

We have now amended the method as below to describe the target population clearly. 

‘The survey was open to participants who were UK-based, of reproductive age from18 years or above, premenopausal and open to women irrespective of the type of contraception they used.’ (Pg 5; Line 144-146)

7) ‘The section could provide more detail on the survey questions and response options. This would help readers understand the specific menstrual cycle characteristics, hormonal symptoms, and lifestyle changes that were assessed in the study.’

This has now been addressed in the methods on (Pg 5-6; Lines 149-156)

8) ‘The section could provide more information on the sample size and characteristics of the study population. This would help readers understand the generalizability of the study findings and the representativeness of the sample.’

We thank the reviewer for this comment. As per reviewer one’s feedback, this has now been addressed in the methods (Pg 4-5; Lines 172-176).

9) ‘Provide more details about the recruitment process. How were the social media advertisements designed? How were they targeted to reach the desired population? Including these details can give readers a better understanding of how participants were selected. Consider mentioning the estimated sample size or the number of participants who took part in the study. This can help readers assess the representativeness and statistical power of the findings.’

We have provided an explanation of how the author advertised the survey through the short stories application on twitter and Instagram, so that readers can better understand how participants were recruited. The sample size has now been described better in the methods section and the results and abstract both state how many participants completed the survey overall. (Pg 5; Lines 135-142)

10) ‘There are a few areas where the discussion could be improved: The section on the impact of the pandemic on menstrual cycles could be more concise. While the study's findings on menstrual cycle changes are important, the discussion could be streamlined to focus on the most significant findings and their implications.’

We appreciate this feedback from the reviewer. We have now reordered some paragraphs to improve flow of the discussion and where possible, tried to make sentences more concise. 

11) The section on the impact of the pandemic on mental health could be expanded. The study's findings on the worsening of PMS symptoms and the high prevalence of anxiety and depression among participants are important, and the discussion could be expanded to explore the potential reasons for these findings and their implications for women's health.

The rationale for the findings have been explained by the HPG and HPA axis theories and because of the changes during the pandemic, including lockdowns and social distancing leading to increased psychological stress, as well as financial insecurity. We refer to increased anxiety and depression placing a financial burden on the NHS, especially if is it not treated prior to pregnancy and post -partum. We have also suggested the implementation of online resources and AI, targeted at women with COVID related anxiety, as successfully demonstrated in countries such as China. (Pg 26; Lines 380-413)

The authors have amended the manuscript as per all reviewers’ feedback, which we feel has strengthened both the arguments discussed and the structure of the manuscript. We hope you find the revised version and answers to the reviewers satisfactory for you to accept this paper in your journal. Please do not hesitate to contact me if we can be of any further assistance in the revision of this document, which will hopefully lead to its final acceptance.

Yours sincerely,

Lorraine Kasaven MBChB BSc (Hons) MRCOG

Clinical Research Fellow

---

## [Editor Report · Decision Letter 1]

8 Aug 2023

The Menstrual Cycle and the COVID-19 Pandemic

PONE-D-23-13834R1

Dear Dr. Lorraine Kasaven,

We’re pleased to inform you that your manuscript has been judged scientifically suitable for publication and will be formally accepted for publication once it meets all outstanding technical requirements.

Kind regards,

Muhammad Junaid Farrukh

Academic Editor

PLOS ONE
---

## [Editor Report · Acceptance letter]

15 Sep 2023

PONE-D-23-13834R1 

The Menstrual Cycle and the COVID-19 Pandemic 

Dear Dr. Kasaven:

I'm pleased to inform you that your manuscript has been deemed suitable for publication in PLOS ONE. Congratulations! Your manuscript is now with our production department. 

Kind regards, 

on behalf of

Dr. Muhammad Junaid Farrukh 

Academic Editor

PLOS ONE